# New Technologies in Digestive Endoscopy for Ulcerative Colitis Patients

**DOI:** 10.3390/biomedicines11082139

**Published:** 2023-07-29

**Authors:** Paolo Biamonte, Ferdinando D’Amico, Ernesto Fasulo, Rukaia Barà, Francesca Bernardi, Mariangela Allocca, Alessandra Zilli, Silvio Danese, Federica Furfaro

**Affiliations:** 1Gastroenterology and Endoscopy, IRCCS San Raffaele Hospital, 20132 Milan, Italy; biamonte.paolo@hsr.it (P.B.); fasulo.ernesto@hsr.it (E.F.); bara.rukaia@hsr.it (R.B.); bernardi.francesca@hsr.it (F.B.); allocca.mariangela@hsr.it (M.A.); zilli.alessandra@hsr.it (A.Z.); danese.silvio@hsr.it (S.D.); furfaro.federica@hsr.it (F.F.); 2Department of Biomedical Sciences, Humanitas University, Pieve Emanuele, 20090 Milan, Italy; 3Gastroenterology and Endoscopy, Vita-Salute San Raffaele University, 20132 Milan, Italy

**Keywords:** ulcerative colitis, endoscopy, technology, artificial intelligence

## Abstract

Ulcerative colitis (UC) is a chronic inflammatory bowel disease primarily affecting the colon and rectum. Endoscopy plays a crucial role in the diagnosis and management of UC. Recent advancements in endoscopic technology, including chromoendoscopy, confocal laser endomicroscopy, endocytoscopy and the use of artificial intelligence, have revolutionized the assessment and treatment of UC patients. These innovative techniques enable early detection of dysplasia and cancer, more precise characterization of disease extent and severity and more targeted biopsies, leading to improved diagnosis and disease monitoring. Furthermore, these advancements have significant implications for therapeutic decision making, empowering clinicians to carefully consider a range of treatment options, including pharmacological therapies, endoscopic interventions and surgical approaches. In this review, we provide an overview of the latest endoscopic technologies and their applications for diagnosing and monitoring UC. We also discuss their impact on treatment decision making, highlighting the potential benefits and limitations of each technique.

## 1. Introduction

Ulcerative colitis (UC) is a chronic inflammatory bowel disease (IBD) characterized by recurrent inflammation of the colon and the rectum [1]. It is associated with substantial morbidity and significantly impacts patients’ quality of life [2]. Disease prevalence is higher in industrialized countries; however, the incidence is also rising in the less developed ones, with an overall incidence of 1.2–20.3 cases per 100,000 persons per year and with a prevalence of 7.6–245 cases per 100,000 per year [3].

The chronic inflammatory stimulus in UC patients is associated with a higher risk of developing colorectal cancer (CRC) as compared to the general population [4]. The reported overall prevalence of CRC in UC patients is 3.7%, with a standardized incidence ratio of 5.7 (95% CI, 4.6–7.0) [5,6].

Endoscopy plays a crucial role throughout the entire course of the disease, starting from the diagnosis. It is indispensable for obtaining histological confirmation, determining the extent and severity of the disease, monitoring activity, performing dysplasia surveillance and providing endoscopic treatment when possible [7,8].

In recent years, new technologies have emerged, which could potentially overcome limitations associated with conventional techniques [9]. These advancements offer the opportunity for ultrastructural, macroscopical and even microscopical evaluation of the disease that were previously unavailable [9].

In this context, the emergence of artificial intelligence (AI) has revolutionized the field of automated image analysis, providing valuable support throughout the management of the disease [10]. 

By exploring the latest advancements in endoscopic technologies, we will uncover the substantial benefits they offer in terms of accurate diagnosis, effective disease monitoring, risk stratification, prediction of disease outcomes and optimization of treatment strategies [11].

While these advancements offer immense potential, their widespread adoption and integration into routine clinical practice require further validation and refinement [11]. Through a critical appraisal of the available evidence, this review aims to provide a comprehensive overview of the current state of endoscopic technologies in the management of UC, specifically in evaluating disease activity and performing endoscopic surveillance. By addressing both the achievements and challenges associated with these technologies, we aim to offer valuable insights to clinicians and researchers in this field. 

## 2. Novel Endoscopic Technologies to Evaluate Disease Activity in UC Patients

### 2.1. Artificial Intelligence (AI)

Endoscopic evaluation plays a critical role in assessing UC activity, by primarily targeting the colonic mucosa and identifying inflammatory changes [12]. The severity of the disease ranges from microscopic alterations without clinical symptoms to visible lesions characterized by erythema, erosions, deep ulcers and spontaneous bleeding [13]. 

To evaluate disease activity during endoscopy, scoring systems such as ulcerative colitis endoscopic index of severity (UCEIS) and Mayo endoscopic score (MES), rely on these visual characteristics [14,15]. However, these existing scores are limited by inter- and intra-observer variability [16,17]. For UCEIS, the estimates of inter-rater and intra-rater reliability shows intraclass correlation coefficients (ICC) of 0.5 and 0.72, respectively [18]. Regarding the MES, inter-rater reliability estimates range between ƙ = 0.45 and 0.75, while intra-rater reliability between ƙ = 0.48 and 0.75—for trainees and experts, respectively [16,19].

In this scenario, AI offers promising solutions by automating the assessment process with increased consistency and objectivity [20]. 

In the field of AI, machine learning (ML) indicates the development of algorithms that enable computers to learn from data and make predictions or decisions without being explicitly programmed [21]. Deep learning (DL) is a subset of machine learning that focuses on the utilization of artificial neural networks with multiple layers, designed to mimic the structure and function of the human brain and enable the model to automatically learn hierarchical representations of data [22]. Neural networks have varying levels of complexity, from simple to more advanced architectures like convolutional neural networks (CNN) [23]. CNNs are advanced network architectures applied in computer vision tasks that use filters to recognize patterns and connect them layer by layer to understand more intricate features [24]. Table 1 summarizes the main studies about AI systems in UC patients.

#### 2.1.1. AI on Still Images

Multiple studies have demonstrated that the implementation of AI in UC management can achieve success rates comparable to human experts when using still images. In 2019, a computer-aided diagnosis device (CADx) was developed based on imaging data of endoscopic examinations from 841 patients, exhibiting high discrimination capability between active disease (MES 2–3) and endoscopic remission (MES 0–1), with an impressive area under the curve value (AUC) of 0.98 (95% CI, 0.97–0.98; *p* < 0.0001) [25]. 

Another study replicated comparable outcomes (AUC 0.97; 95% CI, 0.963–0.969), using a CNN for MES evaluation (MES 0–1 vs. 2–3) and employing over 16,000 images from 3082 patients from the past ten years. Furthermore, the consistency of the MES grade determined by the CNN was found to be equivalent to paired expert reviewer (k = 0.84; 95% CI, 0.83–0.86 vs. k = 0.86; 95% CI, 0.85–0.87) [26]. This may indicate that automated systems can distinguish remission from non-remission based on still images as efficiently as expert humans. 

In another study conducted by Sutton et al. [35], the application of DL algorithms was examined in the context of distinguishing UC from other diseases and in the assessment of the severity of UC, employing the MES as a comparator. The dataset consisted of 851 images, each one previously graded by experienced endoscopists. The most efficient DL model achieved an accuracy of 87.5%, a sensitivity of 79%, a specificity of 91% and an AUC of 0.90.

In a recent meta-analysis including 12 studies investigating CNN algorithms to assess endoscopic severity in UC, AI trained with UCEIS performed better than MES. The sensitivity and positive predictive value of UCEIS were 93.6% (87.5–96.8), I^2^ = 77% compared to 82% (75.6–87), I^2^ = 89% for MES, *p* = 0.003. Similarly, the sensitivity and positive predictive value of MES were 93.6% (88.7–96.4), I^2^ = 68% compared to 83.6% (76.8–88.8), I^2^ = 77% for UCEIS *p* = 0.007 [40].

#### 2.1.2. Real Time Endoscopy

Despite advancements in image analysis, a comprehensive assessment of disease severity during endoscopic examinations requires real-time analysis. However, it presents new challenges, such as patient movements, bowel preparation, peristalsis and biopsy-related mucosal injury, which must be accurately recognized and accounted for by the system. In a recent multicenter trial, full-length high-definition (HD) videos were used to train and validate a CNN to predict MES. The results demonstrated agreement rates of 64% for low-definition videos, 78% for HD videos and 82% when compared to central readers [30]. Gottlieb et al. [29] used 795 full-length endoscopic video data from 95 patients of a mirikizumab phase 2 trial [41] to train a new CNN. This model achieved a high level of agreement with central readers’ MES and UCEIS scores, with weighted kappa measurements of 0.844 (95% CI, 0.787–0.901) and 0.855 (95% CI, 0.80–0.91), respectively. Another study conducted in 2021 evaluated an end-to-end DL CADx system for UC assessment using MES [31]. The AI model was trained on 1672 sigmoidoscopy HD videos obtained from a multicenter dataset originating from three clinical trials (Phase II Eucalyptus, Phase III Hickory and Phase III Laurel trials on etrolizumab). Results showed an AUC of 0.84 for MES ≥ 1; 0.85 for MES ≥ 2 and ≥3. Another CNN model trained on 134 prospective videos was tested for MES and UCEIS prediction during real-time examination [37]. The quadratic weighted kappa used to compare the inter-rater agreement between the labels provided by experts and the predictions made by the model demonstrated a significant level of agreement, indicating a robust concordance (0.78 for UCEIS and 0.9. for MES). Fan et al. [38] developed an interactive AI scoring system to evaluate mucosal inflammation using MES and UCEIS. The AI was trained with 5875 white-light (WL) images and tested on 20 full colonoscopy HD videos from 18 patients with UC, partitioning the large bowel into five macro areas (cecum, ascending, transverse, descending, sigmoid colon, rectum) and multiple sub-segments. This model achieved an accuracy rate of 87% using MES (k = 0.8; CI:95%, 0.782–0.844). When employing UCEIS assessment, accuracy resulted in 90.7% for vascular patterns (k = 0.822; 95% CI, 0.788–0.855), 84.6% for erosions/ulcers (k = 0.784; 95% CI, 0.744–0.823) and 77.7% for bleeding (k = 0.702; 95% CI, 0.612–0.793) evaluation. AI algorithms further analyzed each bowel segment using a standardized system that created a two-dimensional indicator of the severity of sub-segmental inflammation activity. The evolution of AI-systems also led to the development of rapid reading software for video capsule endoscopy, enabling automated detection of lesions and abnormalities and reducing time-consuming evaluation of lengthy video recordings [42]. 

#### 2.1.3. AI for Histological Prediction

The concept of endoscopic remission is expected to evolve towards more in-depth concepts of complete mucosal healing, which indicates the absence of visible inflammation signs and histological healing, which combines both endoscopic and histological remission (HR) [43,44]. Since AI has demonstrated its ability to enhance endoscopic assessment of UC severity to expert standards, exploring its potential in histological assessment is reasonable. 

As histological activity is a known predictor of negative clinical outcomes in UC, the implementation of AI may enable the detection of specific endoscopic features that correlate with histology, potentially reducing or eliminating the need for biopsies, which incur additional expenses. 

Huang et al. [32] utilized a monocentric dataset of 856 full-colonoscopy images from 54 patients with UC categorized by two reviewers into two groups based on their respective MES sub-scores, i.e., 643 images MES 0–1 (mucosal healing) and 213 images MES 2–3 (nonmucosal healing). A pre-trained CNN was employed to train three distinct classifiers: a deep neural network (DNN), a support vector machine (SVM) and a k-nearest neighbor (k-NN) network. The DNN demonstrated an accuracy of 93.8% (sensitivity 84.6% and specificity 96.9%). The SVM reported an accuracy of 94.1% (sensitivity 89.2% and specificity 95.8%), while the k-NN achieved an accuracy of 93.4% (sensitivity 86.2% and specificity 95.8%). The combination of ensemble learning techniques yielded an accuracy of 94.5% (sensitivity 89.2% and specificity 96.3%) in detecting mucosal healing. Furthermore, the system effectively differentiated between MES 0 and MES 1 (i.e., complete mucosal healing), achieving an accuracy of 89.1% (sensitivity 82.3% and specificity 92.2%).

A new automated calculation system based on red density technology, which measures image red pixel metrics and vascularity, was developed by Bossuyt et al. [27] in 2020. Data from a total of 35 patients, 29 with UC and 6 healthy controls, were employed. This approach serves as an alternative to inflammation assessment during endoscopy. The results computed using this technology correlated well with MES (r = 0.76, *p* < 0.0001), UCEIS (r = 0.74, *p* < 0.0001) and Robarts histological index (RHI) (r = 0.74, *p* < 0.0001). 

Takenaka et al. [28] developed a CNN trained on 40,758 colonoscopy images and 6885 corresponding histological analyses to predict HR—defined as a Geboes score < 3.0—with a final diagnostic accuracy of 92.9% accuracy (95% CI 92.1–93.7%) and a kappa coefficient between CNN prediction and biopsy result of 0.859 (95% CI 0.841–0.875).

In the same group’s subsequent prospective study, the CNN prediction of both endoscopic and HR correlated with significantly better clinical outcomes (reduction in steroid use, relapse of disease, hospitalization, colectomy, *p* < 0.001) [34]. 

In an open-label cohort study involving 135 patients with UC in clinical remission, real-time AI-assisted output was used to classify individuals into healing or active group, using the EndoBRAIN-UC (Cybernet Systems Corporation, Tokyo, Japan) [36]. The primary endpoint was the relapse rates for one-year follow-up. Statistical analysis revealed that participants classified as AI-active had significantly higher incidence rates of relapse when compared to those classified as AI-healing (28.4%; 95% CI, 18.5−40.1% vs. 4.9%; 95% CI, 1.0−13.7%; *p* < 0.001).

Iacucci et al. developed a new AI-based tool for endoscopic assessment called Paddington International Virtual ChromoendoScopy ScOre (PICaSSO); it was used in a prospective study in 2021 that involved more than 1000 videos from 283 patients [39]. 

Two separate computer models were developed: one trained on WL and the other on virtual chromoendoscopy (VCE) video data. The VCE-AI system performed better than the WL counterpart to detect endoscopic remission (defined as PICaSSO ≤ 3), reaching a sensitivity of 79%, a specificity of 95% and an AUC of 0.94 (95% CI 0.91–0.97). The AI model demonstrated comparable accuracy in predicting HR in both WL and VCE, with accuracies between 80 and 85%. The agreement between the AI system and endoscopist in detecting disease activity was assessed using Cohen’s kappa coefficient. In VCE videos, k was 0.73, 95% CI 0.61–0.85, while in HD-WLE videos, k was 0.51, 95% CI 0.36–0.66. The agreement between the AI system and pathologist in detecting disease activity was assessed in VCE videos (0.45) and in HD-WLE videos (0.59) [39]. 

In a subsequent study, the PICaSSO system was implemented to detect HR through the PICaSSO histological remission index (PHRI), which utilized an automated dichotomous approach to evaluate neutrophil presence/absence [45]. An AUC of 0.90 (95% CI, 0.86–0.94) for RHI and an AUC of 0.82 (95% CI, 0.75–0.88) for the Nancy histological index (NHI) in detecting PHRI ≤ 3 were registered.

PHRI demonstrated strong associations with MES, UCEIS and PICaSSO. Furthermore, a significant correlation was observed between PHRI and clinical outcomes, such as hospitalization, colectomy and therapy modifications triggered by flare-ups, effectively stratifying patients based on their risk within 12 months. 

Furthermore, a CADx trained with images of 58 patients from a single short wavelength monochromatic LED light illumination endoscope (Fujifilm, Tokyo, Japan) has been developed by Bossuyt et al. [46]. This innovative tool combination allows for the real-time assessment of the mucosal architecture, at a depth ranging 50–200 µm. The algorithm showed a positive predictive value for histologic remission of 0.83, while UCEIS and MES reached, respectively, 0.65 and 0.59.

### 2.2. Confocal Laser Endomicroscopy (CLE) 

In addition to artificial intelligence, other endoscopic techniques are gaining ground and may be applied in UC endoscopic assessment. 

Confocal laser endomicroscopy (CLE) is an advanced endoscopic technique developed to capture highly detailed and magnified images of the mucosa in the gastrointestinal tract. It relies on the illumination of tissue using a low-power laser, followed by the detection of fluorescent light reflected from the tissue through a pinhole after the application of systemic or topical fluorescence agents [33]. Two different systems can be distinguished [47]: Probe-based CLE (pCLE) consists of a bundle of fiber optics with a lens at its distal end, which is connected to a laser scanning unit. On the other hand, endoscope-based CLE (eCLE) involves the integration of a confocal microscope into the distal tip of a conventional endoscope. Applying this technique to detect mucosal healing, Hundorfean et al. developed and validated a specific score for the purpose, the endomicroscopic mucosal healing score (eMHS; range, 0–4) [48]. It showed a sensitivity of 100% (95% CI: 15.8–100%), a specificity of 93.75% (95% CI: 69.8–99.8%) and an accuracy of 94.4%. Furthermore, a strong positive correlation was observed between eMHS and EMS (r = 0.81%, *p*< 0.0001). In addition, eMHS < 1 may predict sustained clinical remission and a reduced need for hospitalization, steroid therapy and surgery.

Another study evaluated 14 morphologic and functional parameters extracted from CLE recordings of 27 UC and 9 control patients to discriminate between IBD and healthy patients and disease activity. A significant increase in the mean intercrypt distance, wall thickness and fluorescein leakage through the mucosa with respective increments of 109%, 117% and 174% (*p* < 0.05) were registered in UC patients when compared to CD ones. The same parameters where increased in IBD individuals when compared with healthy controls [49]. 

### 2.3. Endocytoscopy (EC) 

Another new technique that is taking place in UC evaluation is endocytoscopy (EC) [50]. EC employs a high-powered fixed-focus lens which can be integrated into the endoscope or utilized through a separate probe, enabling magnification of up to ×1440 but does not require administration of intravenous contrast or additional video processors [51]. Several studies have demonstrated the utility of EC for the assessment of disease activity in UC patients. A study conducted by Nakazato et al., focusing on UC patients with ER (i.e., MES 0), showed that EC could accurately distinguish between individuals with histologically active disease and those in HR [52]. Similarly, Iacucci et al. prospectively recruited 29 patients to develop the endocytoscope scoring system (ECSS), which showed a strong correlation with RHI (r = 0.89; 95% CI, 0.51–0.98) and NHI (r = 0.86; 95% CI, 0.42–0.98). However, it correlated poorly with MES (r = 0.28; 95% CI, 0.27–0.70) [53]. Takishima et al., analyzing the results from 120 patients, found that quantification of goblet cells using EC predicted long-term sustained clinical remission in UC patients with ER (i.e., MES 0) [54]. During a median follow-up period of 549 days, patients in the depleted goblet cell group demonstrated a significantly higher cumulative rate of clinical relapse when compared to those in the preserved goblet cell group (19% vs. 5%, respectively; *p* = 0.02). Furthermore, recently, Vitali et al. highlighted the superior accuracy of endocytoscopic assessment in detecting microscopic disease activity in UC as compared to WLE. A new score, the ELECT (ErLangen Endocytoscopy in ColiTis) score, was developed and validated in 46 UC patients. It showed a strong positive correlation with histopathologic scoring systems (RHI r = 0.70 and NHI 0.73) and superiority over WLE when grading microscopic disease activity (sensitivity = 88%, specificity = 95.2% and AUC = 0.916) [55]. Moreover, EC was shown to be as reliable as histology in predicting clinical outcomes for UC patients.

## 3. Endoscopy and UC Surveillance Program

Patients with long-standing UC, excluding those with limited proctitis, have an increased risk of CRC as a result of inflammation-induced carcinogenesis [56]. Recently, the availability of novel medical therapies and the implementation of improved colonoscopic surveillance have led to a reduction in colitis-associated colorectal cancer (CAC) [57]. In fact, CAC cumulative risk after 10, 20 and 30 years of disease has dropped from 2%, 8% and 18% in a 2001 meta-analysis [58] to 1%, 3% and 7% in more recent studies [59]. As non-invasive biomarkers for CAC are lacking, colonoscopy remains the mainstay of CAC risk attenuation. There are no randomized controlled trials comparing surveillance to non-surveillance of CRC incidence and related mortality [60]. However, several observational studies reported that patients with long-standing UC in surveillance colonoscopies have less advanced stages of CRC than UC patients in non-surveillance programs (stage I: 21.5% vs. 21.4%; stage II: 10.5% vs. 20.9%; stage III: 10.5% vs. 20.9%; stage IV: 1.5% vs. 9.7%; *p* < 0.001) [61]. A recent systematic review showed a higher overall incidence of CRC in the non-surveillance vs. surveillance group (3.2% versus 1.8%), with significantly lower CRC-related mortality in the surveillance vs. non-surveillance group (8.5% vs. 22.3%; OR 0.36, 95% CI, 0.19–0.69) [62]. These results are related to a higher rate of early CRC in the surveillance group (15.5% vs. 7.7%; OR 5.40, 95% CI, 1.51–19.30) [60,62]. The first colonoscopy performed in an endoscopic surveillance program of IBD patients is referred to as the “screening colonoscopy”, while subsequent exams are referred to as “surveillance colonoscopy”. Current guidelines recommend the timing of the screening colonoscopy based on the duration of time after symptom onset to be 8–10 years [2]. It is critical to predict individual patient risk to establish specific surveillance intervals (Table 2).

### 3.1. White Light Endoscopy (WLE)

Ideally, screening and surveillance exams should be conducted on patients in remission by experienced endoscopists skilled in detecting neoplasia associated with IBD. Dysplasia in IBD is often flat and challenging to detect. Initially, the recommended screening method was white light endoscopy (WLE) with random four-quadrant biopsies taken every 10 cm [63]. Standard-definition WLE (SD-WLE) is an outdated technology which was found to be inferior to high-definition WLE (HD-WLE) [64]. A retrospective analysis demonstrated that HD-WLE significantly detected more dysplastic lesions than SD-WLE in patients with long-standing UC (32 dysplastic lesions in 24/209 colonoscopies in HD-WLE vs. 11 dysplastic lesions in 8/160 colonoscopies in SD-WLE; *p* < 0.05) [65]. Actually, HD-WLE is the standard of care because a wider visual field and higher pixel density allow better identification and definition of dysplastic/tumor lesions [66].

### 3.2. Chromoendoscopy (CE)

Another available technology is chromoendoscopy (CE), virtual or dye; the latter involves the application of various dyes to the colon’s epithelium to enhance areas of mucosal irregularity and suspected lesion borders [67]. Dye chromoendoscopy (DCE) uses different dye agents, which are divided into absorptive agents (Lugol, methylene blue, toluidine blue and cresyl violet), contrast agents (indigo carmine, acetic acid) and reactive staining agents (congo red, phenol red) [68]. By providing better demarcation of dysplastic lesions, DCE enables targeted biopsies with a reduction in their overall number. The SCENIC consensus recommends performing DCE with indigo carmine or methylene blue because they are the dye agents with the strongest evidence [64] However, despite its widespread use, there are many doubts regarding methylene blue absorption [64]. Several RCT and prospective studies compared DCE with WLE (Table 3). The SCENIC guidelines suggest that DCE should be preferred over WLE when conducting surveillance with standard-definition (SD) or HD colonoscopy [64]. A recent meta-analysis of 10 studies confirmed that DCE was more effective than SD-WLE in identifying dysplasia (risk ratio [RR] 2.12, 95% CI 1.15–3.91) [69]. A prospective surveillance study reported that HD-DCE had a higher detection yield compared to HD-WLE (17/152 vs. 7/153, *p* = 0.032) in patients with long-standing IBD [70]. However, a multicenter prospective randomized clinical trial has shown comparable results in dysplasia detection between HD-DCE and HD-WLE (3.9% in the HD-DCE group vs. 5.6% in the HD-WLE group, *p* = 0.749) [71]. Nonetheless, DCE has several potential drawbacks. The application of dye may not always be evenly distributed across the mucosa, hindering proper visualization. In some cases, the pooling of the dye can further impede accurate examination. Additionally, DCE tends to be more time-consuming when compared to other methods. It is important to note that DCE is not feasible in cases of inadequate colonic cleansing or severe colonic inflammation, as these conditions may exhibit neoplasia-like pit patterns that can result in false-positive findings. Furthermore, it is worth considering that studies evaluating DCE were conducted using standard resolution endoscopes, raising questions about the necessity of dye spraying when employing the newer high-definition endoscopes. VCE is a technique that enables the utilization of contrast enhancement without requiring dye agents. It encompasses various techniques, such as narrow-band imaging (NBI), Fuji intelligent color enhancement (FICE) and i-Scan. These are “push-of-a-button” techniques, quickly available during endoscopic exams [68]. Recent studies showed that VCE (with NBI) performed similarly to DCE and HD-WLE in terms of dysplasia detection (DCE = 21.2% and NBI = 21.5%; OR 1.02, 95% CI 0.44–2.35, *p* = 0.964), but VCE reduced the procedural time by an average of 7 min (*p* < 0.001) [72]. An RCT by Iacucci et al. confirmed that VCE is not inferior to dye-spraying colonoscopy for dysplasia detection during UC surveillance (VCE: 39.1% vs. DCE: 48.1%; *p* = 0.84) [73]. Based on this research, it is evident that the effectiveness of VCE and DCE is comparable [74]. However, considering shorter procedural time and lower costs, VCE is preferred over DCE, especially in high-risk patients [75]. Initially, guideline recommendations suggested performing random biopsies (Rb) during surveillance colonoscopy to detect “invisible lesions”, resulting in a total of 33 biopsies or more. With the spread of novel endoscopic technologies, the recent attitude is to perform targeted biopsies (Tb), obtaining specimens of mucosa abnormalities and resulting in a smaller number of samples. The 2017 ECCO consensus recommended using Tb during CE to detect dysplasia in IBD patients. However, if WLE is used, both Rb and Tb of any visible lesion should be performed [2]. A multicenter RCT showed a similar rate of dysplasia detection between Rb and Tb (9.35% vs. 11.4%; *p* = 0.617) but less examination time in the Tb group vs. the Rb group (26.6 vs. 41.7 min, *p* < 0.001) [76]. Although the Tb approach is cost- and time-effective, the Rb strategy still plays an important role in UC patients with high-risk factors. In fact, in a study of 71 UC patients with coexisting primary sclerosing cholangitis (PSC), neoplasia was detected in 22 colonoscopies by Tb or Rb, with 10 (45.5%) CRC detected only through Rb [77].

### 3.3. Autofluorescence Imaging (AFI)

Autofluorescence imaging (AFI) utilizes the properties of light–tissue interactions to enhance neoplastic tissue. When exposed to ultraviolet or short-wavelength visible light, endogenous fluorophores (collagen, nicotinamide, flavin, porphyrins) emit fluorescence [82]. In FIND-UC randomized controlled trials, DCE vs. AFI have been compared, resulting in equal effectiveness for neoplasia detection in patients with longstanding UC (12% using AFI and 19% using CE) [83]. However, current data about AFI are missing to evaluate its role in clinical practice.

### 3.4. Confocal Laser Endomicroscopy (CLE) and Endocytoscopy (EC)

The latest endoscopic techniques, such as confocal laser endomicroscopy (CLE) and endocytoscopy (EC), enable the endoscopist to visualize structures at a subcellular level (e.g., crypt, capillaries, nucleus and cytoplasm), obtaining histologic real time in vivo imaging [68]. CLE emits a low-power laser onto the tissue which is reflected from the tissue and refocused onto the detection system. Thus, systemically (fluorescein sodium) or topically (acriflavine hydrochloride, cresyl violet) exogenous fluorescence agents need to be applied to obtain a confocal image, which is microscopic imaging at 1000-fold magnification in real-time [33]. In 2007, an RCT conducted on 153 UC patients showed that CLE detected more neoplasia than convectional colonoscopy by performing half of the biopsies (19/80 vs. 4/73; *p* = 0.005) [80]. Moreover, a meta-analysis evaluated that CLE sensitivity and specificity were 91% and 97%, respectively, resulting in a highly accurate technology for discerning the neoplastic lesions from non-neoplastic lesions [84]. Application of molecular imaging, such as CLE and fluoroscopy, may help to detect dysplasia and cancer in UC patients. Mitsunaga et al. conducted a study using a mouse model of CAC to investigate the expression of gamma-glutamyltranspeptidase (GGT), an enzyme associated with malignancy. They employed an enzymatically activatable probe called gGlu-HMRG, which was topically applied. Remarkably, within just 5 min, fluorescent areas were detected using this probe, and subsequent pathological examination confirmed that all these areas contained dysplasia or cancer [85]. This finding suggests that molecular imaging, once further developed, could serve as a valuable technique for cancer surveillance and early diagnosis in patients with IBD. However, the routinary application of CLE in surveillance programs is limited by high cost, long program time and need for additional equipment [63]. EC is an advanced imaging technology based on light microscopy with a fixed-focus and high-power objective lens that allows in vivo assessment of GI lesions with an ultramagnification of pathological tissues (up to 1390-fold) [68]. Kudo’s team developed an EC classification system to distinguish between neoplastic and non-neoplastic lesions [86]. In comparison to conventional endoscopy, EC demonstrated significantly higher diagnostic accuracy in predicting both neoplastic and non-neoplastic lesions but in non-UC patients (*p* = 0.015) [87]. More recently, they conducted further research on predicting UC-associated neoplasia (UCAN) using two strategies: pit pattern (PIT) alone or a combination of EC-irregular nuclei to PIT (EC-IN-PIT). EC-IN-PIT improved the predictive values, particularly in terms of specificity (84% vs. 58%, *p* < 0.001) and accuracy (88% vs. 67%, *p* < 0.01), compared to PIT alone [87]. Although this represents the first evidence of the clinical impact of EC nuclear irregularities in predicting UCAN, EC’s role in UC surveillance still requires further discussion.

### 3.5. Artificial Intelligence (AI)

A CAD system called EndoBrain^®^ has been successfully developed to distinguish adenoma from malignant lesions observed by EC in non-IBD patients [88]. In UC patients with dysplasia, data are still missing, but recently, Fukunada et al. published a case study encouraging the application of this technique [89]. Nowadays, there are not enough proven data on the use of AI in surveillance and diagnosis of CAC. The only evidence on the use of AI in CAC detection was a case report, redacted by Maeda et al. [90]. An even more extended application of AI involves its utilization in the prediction of p53 mutations in UCAN or dysplasia, as studied by Noguchi et al. [91]. They employed a DL-CNN model that showed a sensitivity, specificity and precision of 82%, 89% and 77%, respectively. As a result, this AI model could serve as a dependable alternative to p53 immunohistochemistry staining, offering both time and cost savings [91]. More studies should be conducted to bring AI into clinical practice.

## 4. Discussion

The emergence of new technologies in endoscopy has led to significant advancements in UC diagnosis and management. In this review, we have highlighted the potential benefits of CE, CLE, EC and AI in improving the assessment and treatment of UC patients. In particular, CE offers improved visualization of subtle changes in the colonic mucosa, allowing for the early detection of dysplasia and cancer [67]. By enhancing the visibility of abnormal tissue, CE enables more accurate characterization of disease extent and severity, which is crucial for appropriate treatment decisions [68]. This technique can aid in targeting biopsies and guiding therapeutic interventions, leading to better patient outcomes [67]. CLE yields real-time high-resolution images of the gastrointestinal mucosa at a microscopic level [33]. This technology allows endoscopists to evaluate disease activity, assessing mucosal inflammation. To distinguish between inflamed and non-inflamed areas can help in optimizing treatment strategies and monitoring responses to therapy [33]. EC helps visualize individual cells and their characteristics and helps differentiate between neoplastic and non-neoplastic lesions [87]. The AI can analyze endoscopic images and videos and assist endoscopists in real-time decision making [10]. The new AI-based systems can detect and classify abnormalities, predict disease progression and suggest when to optimize treatment in UC patients [21]. While these new technologies offer exciting possibilities, it is important to acknowledge their limitations. Each technique has its own learning curve and may require specialized training for proper implementation [11]. Additionally, cost considerations and accessibility to these advanced technologies may vary across healthcare settings [23]. Further research is needed to validate their clinical utility, establish standardized protocols and evaluate their long-term impact on patient outcomes. In conclusion, the introduction of new technologies in digestive endoscopy has revolutionized the assessment and treatment of UC. CE, CLE, EC and AI-based systems have the potential to improve diagnostic accuracy, disease monitoring and treatment decision making [68]. These innovations provide clinicians with valuable tools to enhance personalized patient care and optimize therapeutic approaches. Due to the recent deployment of these new tools, at the moment there are no comparative studies to determine which one may be preferable in specific clinical settings. Even today, the most well-established techniques, such as CE, are more implemented in daily practice when compared to the others, but in the future, this situation could drastically change. Continued research and advancements in these technologies will further shape the future of UC management and will improve outcomes for patients.

## 5. Conclusions

In the current era of the treat-to-target approach, endoscopy plays an important role in the management of UC patients. The advanced endoscopic technologies displayed through this review represent a reliable and efficient aid in each step of the course of the disease. Undoubtedly, these tools offer substantial benefits in clinical practice by facilitating standardization during endoscopic exams. Implementation of these technologies can result in significant savings in time, costs, effort and human resources. Although substantial investment might be needed for their integration into daily practice, current data show positive large-scale cost-effectiveness. Moreover, modern endoscopic technologies allow a deeper characterization of the disease, bringing an easier and more tailored therapeutic choice. AI-models may predict UC evolution to avoid potential disease flare-ups in the short-medium term. Soon, AI-based prognostic models will probably encompass the integration of endoscopic, clinical, histological, laboratory and imaging data as well as transcriptional, proteomic and microbiome biomarkers. Despite their promising ability to analyze and integrate plenty of data, currently, these new techniques are not commonly used in clinical practice because of a lack of standardization. It is also relevant to mention that these advanced models offer a prediction rather than unequivocal conclusions, with ethical implications arising in their application. Recently, the first efforts of guidance for clinical trial reporting on AI-model development and clinical usefulness have been taken by the Consolidated Standards of Reporting Trials–Artificial Intelligence (CONSORT-AI) and Standard Protocol Items: Recommendations for Interventional Trials–Artificial Intelligence (SPIRIT-AI) and also by the European Society of Gastrointestinal Endoscopy (ESGE) [92,93,94]. Prospective, multidisciplinary and multicenter studies are required to establish each potential use of new instruments in UC management.

## Figures and Tables

**Table 1 biomedicines-11-02139-t001:** Summary of available data on AI-based tools for endoscopic evaluation of UC.

Study (Year)	Design	Outcome	Sample	Imaging Type	Results
Ozawa et al. (2019) [25]	Retrospective	MES 0 and MES 0–1	26.304 images; 841 patients	Still images; colonoscopy	MES 0: AUC 0.86 MES 0–1: AUC 0.98
Stidham et al. (2019) [26]	Retrospective	MES 0–1 vs. MES 2–3	images; 2778 patients	Still images + videos; colonoscopy	Sn 83% Sp 96%
Bossuyt et al. (2020) [27]	Prospective	Red density score	29 patients	Prototype endoscopy	MES: r = 0.79 UCEIS: r = 0.69 RHI: r = 0.60
Takenaka et al. (2020) [28]	Prospective	HR prediction	40.758 images + 6885 biopsies; 2012 patients	Still images; colonoscopy	Acc 90.1% (vs. endoscopists)Acc 92.1% (vs. histology)
Gottlieb et al. (2021) [29]	Prospective	MES and UCEIS prediction	636 videos; 249 patients	Videos; colonoscopy	UCEIS QWK: 0.844 MES QWK: 0.855
Yao et al. (2021) [30]	Prospective	MES prediction	51 videos; 51 patients	Videos; colonoscopy	Sn90% Sp 87%
Gutierrez Becker et al. (2021) [31]	Prospective	MES prediction	1672 videos; 3 RCTs	Videos + Images; colonoscopy	MES ≥ 1; ≥:2; ≥:3: AUC 0.84; 0.85; 0.85
Huang et al. (2021) [32]	Retrospective	MES 0–1 vs. MES 2–3 MES 0 vs. MES 1	600 images; 452 images; Pre-set database	Still images; colonoscopy	MES 0: 89% Acc MES 0–1: 94.5% Acc
Bossuyt et al. (2021) [33]	Prospective	HR prediction	Images from 113 colonic segments; 58 patients	Prototype endoscopy	PPV: 0.83
Takenaka et al. (2021) [34]	Prospective	Clinical outcome prediction	26.250 images; 875 patients	Still images; colonoscopy	PPV: 86.2% NPV: 95.1%
Sutton et al. (2022) [35]	Retrospective	MES prediction	681 images from public dataset	Still images; colonoscopy	Acc: 87.5% AUC: 0.9
Maeda et al. (2022) [36]	Prospective	MES > 2 prediction 1 y after colonoscopy	145 patients	Videos; colonoscopy	28.4% active group vs 4.9% healing group
Byrne et al. (2023) [37]	Prospective	MES and UCEIS prediction	134 videos; 1550.030 images	Videos; colonoscopy	MES QWK: 0.9 UCEIS QWK: 0.78
Fan et al. (2023) [38]	Retrospective	MES and UCEIS prediction	5875 images 20 videos; 332 patients	Videos+ images; colonoscopy	MES: Acc 87% UCEIS (vascular pattern, erosions, bleeding,): Acc 90.7%, 84.6%, 77.7%
Iacucci et al. (2023) [39]	Prospective	ER prediction in WLE and VCE	1090 videos; images; 283 patients	Videos; colonoscopy	ER WLE: AUC 0.85 ER VCE: AUC 0.95

MES, Mayo endoscopic score; UCEIS, ulcerative colitis endoscopic index of severity; HR, histological remission; ER, endoscopic remission; RHI, Robarts histological index; WLE, white light endoscopy; AUC, area under the curve; Sn, sensitivity; Sp, specificity; Acc, accuracy; PPV, positive predictive value; NPV negative predictive value; QWK, quadratic weighted kappa.

**Table 2 biomedicines-11-02139-t002:** Interval surveillance based on individual patient risk in CU longstanding.

	High Risk	Intermediate Risk	Low Risk
Definition	FDRs with CRC < 50 years old PSC Previous dysplasia diagnosis Extensive colitis with severe inflammation	FDRs with CRC > 50 years old Mildly active endoscopic/histologic inflammation	Left-sided colitis without active inflammation Disease remission since the last colonoscopy 2 consecutive exams without dysplasia
Interval surveillance	Annual	Every 2–3 years	Every 5 years

CRC, colorectal cancer; PSC, primary sclerosing cholangitis; FDRs, first-degree relatives.

**Table 3 biomedicines-11-02139-t003:** Dye chromoendoscopy vs. white light endoscopy in UC patient surveillance.

Study (Year)	Design	Patients (*n*)	Dysplasia Detection DCE vs. WLE (Number)	*p* Value
Kiesslich et al. (2003) [78]	Randomized 1:1 DCE with MB vs. WLE	165	32 vs. 10	0.001
Hurlstone et al. (2005) [79]	Prospective DCE with IC vs. WLE	700	69 vs. 24	0.001
Kiesslich et al. (2007) [80]	Randomized 1:1 DCE with MB vs. WLE	153	19 vs. 4	0.005
Gunther et al. (2011) [81]	Randomized 1:1 DCE with IC vs. WLE	150	6 vs. 0	0.005
Yang et al. (2019) [71]	Randomized 1:1 DCE vs. WLE	210	4 vs. 6	0.749
Alexandersson et al. (2020) [70]	Prospective DCE with IC vs. WLE	305	17 vs. 7	0.032

DCE, dye chromoendoscopy; WLE, white light endoscopy; MB, methylene blue; IC, indigo carmine.

## Data Availability

Not applicable.

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
