# Peer review of "New Technologies in Digestive Endoscopy for Ulcerative Colitis Patients"

_biomedicines, 2023, doi:10.3390/biomedicines11082139_

Round 1
Reviewer 1 Report
This is a well written review of new endoscopic technologies for the monitoring of disease activity and detection of premalignant lesions in ulcerative colitis. I only have a few comments:
The review is in part extensive and may overburden the non-gastroenterologist specialist. Some information on the technical details may be shortened since the tables also give an excellent overview. Adding some subheadings would give the manuscript a clearer structure. Eg. AI on still images and real time endoscopy. Some information on AI for the detection of premalignant lesions should be added. In return some redundant information (eg. paragraph 3 and 4 on colorectal lesions) should be shortened and merged.
Author Response
Response to the editor(s) of ‘Biomedicines’ [biomedicines-2531257]
Dear Editor(s),
We sincerely thank you for giving us the opportunity to submit a revised version of our manuscript entitled “New technologies in digestive endoscopy for ulcerative colitis patients" by Biamonte et al.
We kindly thank the Reviewers for the precious comments. We are pleased to know that you appreciated the topic of our manuscript. The manuscript has been significantly revised and improved according to the received suggestions. Included below you can find a point-by-point response to the remarks.
Sincerely,
Federica Furfaro, MD, PhD.
Rev 1
- “This is a well written review of new technologies for the monitoring of disease activity and detection of premalignant lesions in ulcerative colitis”.
Reply: We thank you for your appreciation.
- “Some information on the technical details may be shortened since the tables also give an excellent overview”.
Reply: Thank you for your comment. We made the appropriate changes as recommended.
- "Adding some subheadings would give the manuscript a clearer structure”.
Reply: We have appreciated the reviewer’s comment and we have made the appropriate changes.
- "Some information on AI for the detection of premalignant lesions should be added”.
Reply: We thank the reviewer for this comment. We made the recommended changes. There are very few data on this topic.
- "In return some redundant information (e. paragraph 3 and 4 on colorectal lesions) should be shortened and merged”.
Reply: We made the appropriate changes.

Reviewer 2 Report
This review paper covers new technologies for the detection and treatment of ulcerative colitis. There are three technologies focused here, namely AI, confocal laser endomicroscopy, and endocytoscopy. Overall the review topic is new and insightful. The manuscript is very nicely written and easy to follow.
There are some minor suggestions and comments:
1) Sections 2.2 and 2.3 on the other two techniques are shorter than the coverage on AI. This may be fine considering the potential scope of AI compared to the other two techniques. This is just an observation.
2) In the technologies reviewed, how is the inter- and intra user variability reduced. Are there any suggestions to reduce this variability based on what others have done?
3) As mentioned by the authors, the AI technologies help tremendously with the rapid reading of recorded endoscopy videos. Could you add a few sentences on rapid reading software technologies are beneficial, especially with wireless capsule endoscopes. Please refer to a recent review covers this topic that provides examples of rapid reading software: Miley et. al., "Video Capsule Endoscopy and Ingestible Electronics: Emerging Trends in Sensors, Circuits, Materials, Telemetry, Optics, and Rapid Reading Software", 2021, https://spj.science.org/doi/10.34133/2021/9854040
4) In Section 3, a number of lighting conditions are discussed, such as white light endoscopy, dye-chromoendoscopy, narrow band imaging etc. Does the use of AI automated detection programs work with all lighting conditions or is it trained for one type of video recording in the literature you surveyed?
5) In the Discussion, is there a preference for any specific type of technology for a certain condition or intended application? Some sort of comparative analysis between the techniques would be useful for readers.
Author Response
Response to the editor(s) of ‘Biomedicines’ [biomedicines-2531257]
Dear Editor(s),
We sincerely thank you for giving us the opportunity to submit a revised version of our manuscript entitled “New technologies in digestive endoscopy for ulcerative colitis patients" by Biamonte et al.
We kindly thank the Reviewers for the precious comments. We are pleased to know that you appreciated the topic of our manuscript. The manuscript has been significantly revised and improved according to the received suggestions. Included below you can find a point-by-point response to the remarks.
Sincerely,
Federica Furfaro, MD, PhD.
Rev 2
- “Overall the review topic is new and insightful. The manuscript is very nicely written and easy to follow”.
Reply: We thank you for your appreciation.
- “1) Sections 2.2 and 2.3 on the other two techniques are shorter than the coverage on AI. This may be fine considering the potential scope of AI compared to the other two techniques. This is just an observation”.
Reply: We thank you for your observation.
- “2) In the technologies reviewed, how is the inter- and intra user variability reduced. Are there any suggestions to reduce this variability based on what others have done?”
Reply: We thank you for your suggestion. We added the agreement between AI and endoscopists/pathologists in the text.
- “3) As mentioned by the authors, the AI technologies help tremendously with the rapid reading of recorded endoscopy videos. Could you add a few sentences on rapid reading software technologies are beneficial, especially with wireless capsule endoscopes. Please refer to a recent review covers this topic that provides examples of rapid reading software: Miley et al., “Video capsule Endoscopy and Ingestible Electronics: Emerging trends in sensors, circuits, materials, telemetry, optics, and rapid reading software” 2021”.
Reply: We thank you for your suggestion. We have added this information, as requested.
- “4) In Section 3, a number of lighting conditions are discussed, such as white light endoscopy, dye-chromoendoscopy, narrow band imaging etc. Does the use of AI automated detection programs work with all lighting conditions or is it trained for one type of video recording in the literature you surveyed?”.
Reply: In Section 2, it is explained how AI was trained for disease activity assessment using both white light endoscopy (WLE) and video capsule endoscopy (VCE). However, in Section 3, such information was not reported because there were no data available on the use of AI in IBD surveillance.
- “5) In the discussion is there a preference for any specific type of technology for a certain condition or intended application? Some sort of comparative analysis between the techniques would be useful for readers”.
Reply: We thank for this comment. We added a comment about that in the discussion
